# Loop Dynamics Mediate Thermal Adaptation of Two Xylanases from Marine Bacteria

**DOI:** 10.3390/ijms26073215

**Published:** 2025-03-30

**Authors:** Jinhua Zhuang, Yuxi Zhang, Yawei Wang, Zhenggang Han, Jiangke Yang

**Affiliations:** College of Life Science and Technology, Wuhan Polytechnic University, Wuhan 430023, China; 15986892458@163.com (J.Z.); zhangyuxi162@163.com (Y.Z.); wyw860109@126.com (Y.W.)

**Keywords:** low-temperature enzymes, xylanases, glycoside hydrolase family 10, molecular dynamics simulation

## Abstract

This study investigates the biochemical properties of two xylanases, ZgXyn10A and CaXyn10B, which are members of the glycoside hydrolase family 10 (GH10) and originate from the marine Bacteroidetes species *Zobellia galactanivorans* and *Cellulophaga algicola*, respectively. Utilizing an auto-induction expression system in *Escherichia coli*, high-purity recombinant forms of these enzymes were successfully produced. Biochemical assays revealed that ZgXyn10A and CaXyn10B exhibit optimal activities at 40 °C and 30 °C, respectively, and demonstrate a high sensitivity to temperature fluctuations. Unlike conventional low-temperature enzymes, these xylanases retain only a fraction of their maximal activity at lower temperatures. To gain deeper insights into the structural and functional properties of these marine xylanases, two thermostable GH10 xylanases, TmxB and CoXyn10A, which share comparable amino acid sequence identity with ZgXyn10A and CaXyn10B, were selected for structural comparison. All four marine xylanases share a nearly similar three-dimensional structural topology. Molecular dynamics simulation indicated a striking difference in structural fluctuations between the low-temperature and thermostable xylanases, as evidenced by the distinct root mean square deviation values. Moreover, root mean square fluctuation analysis specifically identified the β3-α3 and β7-α7 loop regions within the substrate-binding cleft as crucial determinants of the temperature characteristics of these GH10 xylanases. Our findings establish loop dynamics as a key evolutionary driver in the thermal adaptation of GH10 xylanases and propose a loop engineering strategy for the development of industrial biocatalysts with tailored temperature responses, particularly for lignocellulosic biomass processing under moderate thermal conditions.

## 1. Introduction

Xylanases, enzymes that hydrolyze 1,4-β-D-xylosidic linkages in xylan, a major structural polysaccharide in plant cells or a major component of hemicellulose, have garnered significant attention in biotechnology due to their diverse industrial applications. From biofuel production to food and pulp processing, these enzymes enhance efficiency and sustainability by enabling the breakdown of lignocellulosic biomass under mild conditions [1]. Xylanases are classified into glycoside hydrolase (GH) families based on the sequence and catalytic mechanism, with GH10 enzymes standing out for their broad substrate specificity and robust activity across diverse xylan structures [2]. However, tailoring enzymatic properties, such as the thermal stability and low-temperature activity, remains a critical challenge for optimizing industrial processes [3].

Enzymes with distinct thermal adaptations, ranging from cold-active variants operating at temperatures below 25 °C to thermostable counterparts functional at 55 °C or above, hold complementary industrial significance [4]. Cold-active enzymes, characterized by high catalytic efficiency at low temperatures, offer advantages in terms of the energy savings, reduced contamination risks, and ease of inactivation. These properties make them ideal for applications like food processing, textile manufacturing, and bioremediation [5]. Conversely, thermostable xylanases exhibit remarkable stability under extreme heat, enabling their use in energy-intensive processes such as lignocellulosic biomass pretreatment and pulp bleaching, where high temperatures enhance the substrate solubility and reaction kinetics while minimizing the microbial contamination [6]. It is necessary to decode the temperature-specific structural determinants mediating thermal adaptation in xylanases.

Marine environments, with their extreme conditions and biodiversity, are rich sources of novel enzymes. Marine bacteria have evolved unique enzymatic repertoires to degrade complex algal polysaccharides under cold, saline conditions, which makes them attractive candidates for industrial applications [7]. Many useful enzymes have been discovered from marine samples through the metagenomic approach [8]. *Zobellia galactanivorans* is a model organism to study the degradation of algal polysaccharides [9,10,11]. *Cellulophaga algicola* produces a wide range of extracellular enzymes capable of degrading algal polysaccharides at low temperatures (20–30 °C) [12,13,14]. Only one GH5 glucanase involved in the biodegradation of land plant polysaccharides was identified in *Z. galactanivorans* and *C. algicola*, respectively [15,16]. GH10 xylanases involved in the degradation of plant hemicelluloses have not been characterized from *Z. galactanivorans* or *C. algicola*.

In this study, we investigate two GH10 xylanases, ZgXyn10A from *Z. galactanivorans* and CaXyn10B from *C. algicola*, which exhibit optimal activities at 40 °C and 30 °C, respectively. Unlike typical cold-active enzymes, both show significantly reduced activity at lower temperatures, suggesting unique structural adaptations. To elucidate the molecular basis of their thermal behavior, we compare their structures and dynamics with two thermostable marine GH10 xylanases, TmxB [17] and CoXyn10A [18], using molecular dynamics (MD) simulations. Despite these four xylanases sharing similar amino acid sequence and structural identities, the MD analyses reveal critical differences in loop flexibility and substrate-binding cleft dynamics that correlate with thermal stability. These findings advance our understanding of GH10 xylanase evolution and provide a framework for engineering enzymes with tailored thermal properties for industrial use.

## 2. Results

### 2.1. Sequence and Structural Features of ZgXyn10A and CaXyn10B

The ZgXyn10A gene encodes a 379-amino-acid protein containing a 17-residue signal peptide (NCBI sequence accession number: WP_013991550.1), while CaXyn10B encodes a 398-amino-acid protein with a 24-residue signal peptide (NCBI sequence accession number: WP_013549131.1). Both proteins possess a single catalytic domain adopting the canonical (β/α)_8_ barrel fold characteristic of the GH10 family xylanases, lacking auxiliary domains. Multiple sequence alignment of their catalytic domains with the thermophilic homologs TmxB and CoXyn10A revealed moderate sequence identities ranging from 27% to 42% (Figure 1, Table 1).

The structural superposition of the Cα atoms revealed similar spatial configurations (Figure 2 and Appendix A). ZgXyn10A and CaXyn10B exhibited a root mean square deviation (RMSD) of 1.56 Å. Comparisons with the thermophilic homologs showed similar deviations, ranging from 1.25 to 1.78 Å.

### 2.2. Enzymatic Properties

Both enzymes were expressed in *E. coli* and purified to high concentrations. The purified proteins migrated as single bands on SDS-PAGE, corresponding to the theoretical molecular weights (ZgXyn10A: 43.7 kDa; CaXyn10B: 44.8 kDa) (Figure 3). The protein concentrations determined by the Bradford assay were 4.4 mg/mL and 4.5 mg/mL for ZgXyn10A and CaXyn10B, respectively.

The enzymatic assays revealed different pH and temperature profiles for ZgXyn10A and CaXyn10B (Figure 4A,B). Both enzymes showed optimal activity at pH 6–7. ZgXyn10A maintained 90% maximal activity at pH 5 compared to CaXyn10B’s near-complete inactivation. The temperature optima differed significantly: 40 °C for ZgXyn10A and 30 °C for CaXyn10B. At 20 °C, ZgXyn10A retained 30% activity versus CaXyn10B’s 50%, decreasing to 10% and 23%, respectively, at 10 °C.

The stability analyses demonstrated superior pH tolerance in ZgXyn10A, retaining no less than 70% activity across pH 4–11, versus CaXyn10B’s narrower stability range (pH 5–10) (Figure 4C). The thermal stability assays showed that ZgXyn10A maintained full activity after 1 h at 30 °C, while CaXyn10B lost 40% activity. At 40 °C, ZgXyn10A retained 40% activity compared to complete inactivation for CaXyn10B (Figure 4D). Both enzymes exhibited exceptional halotolerance, maintaining full activity in 0–2.5 M NaCl (Figure 4E).

The metal ion effects differed markedly. CaXyn10B’s activity was enhanced 1.4–1.8-fold by Ca^2^⁺, Co^2^⁺, Mg^2^⁺, and Mn^2^⁺, while ZgXyn10A remained largely unaffected (Figure 4F). When EDTA was added, CaXyn10B’s activity decreased more (by 35%) compared to ZgXyn10A (only 15% reduction). Both enzymes were inhibited by SDS, β-mercaptoethanol, isopropanol, and acetone, but they were not inhibited by non-ionic detergents, urea, or common organic solvents.

Kinetic analysis using the Michaelis–Menten model revealed ZgXyn10A (*k*_cat_/*K*_m_ = 115.2 mL/s/mg) to be much more efficient compared to CaXyn10B (10.23 mL/s/mg), driven by both the lower *K*_m_ (8.84 vs. 16.00 mg/mL) and higher *k*_cat_ (1017.65 vs. 163.69 s^−1^) (Table 2) (Appendix A).

### 2.3. Loop Dynamics in Substrate-Bonding Cleft Are Determinants of Thermos Properties

The 100 ns molecular dynamics (MD) simulations revealed distinct flexibility patterns among the enzymes ZgXyn10A, CaXyn10B, TmxB, and CoXyn10A (Figure 5 and Appendix A). Specifically, TmxB and CoXyn10A demonstrated root mean square deviation (RMSD) values below 0.15 nm and root mean square fluctuation (RMSF) peak values for the β7-α7 loop below 0.15 nm. In contrast, ZgXyn10A and CaXyn10B exhibited higher RMSD values (>0.15 nm) and elevated loop RMSF values (>0.15 nm). Notably, the marine enzymes ZgXyn10A and CaXyn10B displayed greater RMSD fluctuations compared to their thermophilic homologs, TmxB and CoXyn10A, with CaXyn10B showing the most significant structural deviation. During the simulations at different temperatures, the backbone RMSD of ZgXyn10A stabilized at different values (being the highest RMSD at 315 K), while that of CaXyn10B fluctuated around the same value for all three temperatures (Appendix A). In contrast, both TmxB and CoXyn10A maintained relatively stable RMSD values without significant temperature-related changes at all three temperature settings.

The RMSF analysis further highlighted critical differences in the flexibility of the substrate-binding cleft loops among the enzymes (Figure 6). While the (β/α)_8_ barrel cores remained stable, the β3-α3 and β7-α7 loop residues exhibited pronounced flexibility in CaXyn10B. ZgXyn10A also displayed additional mobility in the β2-α2 loop. Conversely, the thermophilic homologs maintained rigidity in these regions, except for the moderate flexibility observed in the β3-α3 loop. It is worth noting that the β7-α7 loops are adjacent to the catalytic residues (general acid/base: Glu160 in ZgXyn10A and Glu177 in CaXyn10B; catalytic nucleophile: Glu265 in ZgXyn10A and Glu284 in CaXyn10B), underscoring the functional significance of their observed flexibility patterns.

## 3. Discussion

The bioprospecting of extremophilic xylanases remains crucial for industrial applications requiring enzymatic stability under harsh conditions [19]. While thermostable xylanases dominate biotechnological applications (e.g., pulp bleaching at 60–90 °C) [20], psychrotolerant variants hold promise for energy-efficient processes at ambient temperatures [6]. Marine ecosystems, characterized by fluctuating temperatures and high salinity, represent an underexplored reservoir of enzymes with unique adaptations [7]. Our identification of ZgXyn10A and CaXyn10B from marine bacteria aligns with recent reports highlighting marine microbial xylanases with atypical thermal profiles [21,22].

Both xylanases exhibited exceptional halotolerance (full activity at 2.5 M NaCl), surpassing the typical marine/soda lake GH10 counterparts from *T. maritima* MSB8 (TmxB) [17], *Thermoanaerobacterium saccharolyticum* NTOU1 [23] and XynSL4 from *Planococcus* sp. SL4 [24], *Bacillus* sp. SN5 [25], which retained 65%, 71%, 70% (approximately) and 90% activity in 2 M NaCl, respectively. The thermal profiles of ZgXyn10A (T_opt_ 40 °C) and CaXyn10B (T_opt_ 30 °C) are different. Compared to ZgXyn10A, CaXyn10B is more like a cold-active enzyme, as demonstrated by the lower optimal temperature and larger *K*_m_ value [26]. However, unlike typical cold-active enzymes, which maintain a significantly higher percentage of their optimal activity between 0 and 10 °C, both enzymes studied here showed reduced performance. They retained only 10–50% residual activity even in the relatively higher temperature range of 10–20 °C (Figure 4) [26].

Our discovery strategy leveraged a sequence identity search targeting ~30% identity with the hyperthermostable GH10 xylanases TmxB and CoXyn10A [17]. This threshold balances functional conservation (retaining catalytic residues) with structural divergence enabling novel properties. As demonstrated by ZgXyn10A (30% identity to TmxB) and CaXyn10B (27%), even limited sequence identity preserves the GH10 catalytic scaffold while permitting loop modifications critical for substrate accommodation (Figure 1 and Figure 2). Recent advances in AI-driven enzyme prediction (e.g., AlphaFold2 [27]) could refine such strategies by predicting flexibility hotspots beyond sequence alignment. However, our results validate traditional homology-based screening as effective for initial enzyme triaging, particularly when complemented by structural simulations.

The comparative MD simulations exposed fundamental differences between marine and thermophilic GH10 xylanases. While the thermophiles TmxB and CoXyn10A maintained low RMSD (<0.15 nm) throughout the simulations, the marine variants showed elevated fluctuations (Figure 5). The simulations at 285 K, 300 K, and 315 K further confirmed the difference in the RMSD between the two xylanase types. The thermophilic xylanases showed stronger resistance to temperature changes compared to the cold-adapted ones (Appendix A). Although a difference of 0.1–0.2 nm in length is not considered significant in protein three-dimensional structural comparisons, all four xylanase molecules consistently maintained this disparity throughout nearly the entire 100 ns simulation period. The RMSF analysis localized the differences to the loops that connect each pair of β-strands and α-helices (Figure 1 and Figure 2). These loops form the substrate-binding cleft on the surface of the GH10 domain. The RMSF values indicated that the β3-α3 loop and β7-α7 loop of ZgXyn10A and CaXyn10B are much more flexible than those of TmxB and CoXyn10A. Especially, the difference in the β7-α7 loop is the most significant (Figure 6). As the temperature increases, the flexibility of these loops in ZgXyn10A and CaXyn10B exhibits a slight upward trend (Appendix A). However, the increase is very small. Whether it is significant needs to be tested with more simulations over a wider temperature range in the future. Since these loops are very important for substrate binding and catalysis (and the catalytic triad is also located on these loops), our findings back up the conventional understanding that rigid loops are crucial for the thermostability of enzymes, as they can stabilize the substrate-binding cleft against thermal denaturation of the GH10 domain [28].

Contrary to expectations, the hydrogen bond density exhibited no correlation with the thermal stability (Appendix A), indicating that global structural rigidity plays a lesser role than local conformational control. Based on our findings, preliminary correlations between GH10 family xylanase structures and temperature-dependent characteristics were identified: those with high RMSD and high loop RMSF tend to favor mesophilic/psychrotolerant activity. This observation provides an empirical approach for predicting enzyme thermostability using short MD trajectories.

In conclusion, our integrative analysis reveals that moderate sequence similarity (~30%) to thermophilic templates insufficiently predicts thermal adaptation in GH10 xylanases, as local structural dynamics in the loop regions override global sequence–structure relationships. Marine-derived ZgXyn10A and CaXyn10B exemplify the evolutionary complexity of enzyme adaptation. The identified loop flexibility signatures provide a roadmap for rational engineering of GH10 xylanases, enabling targeted modulation of the thermal profiles without compromising the catalytic efficiency. However, further analysis of the loop region dynamics in relation to the temperature characteristics requires examining a larger dataset of GH10 family xylanases with completed enzymatic characterization. Future studies should expand this approach to diverse GH families, leveraging machine learning to correlate simulation metrics with experimental stability data.

## 4. Materials and Methods

### 4.1. Materials

The sequences of ZgXyn10A and CaXyn10B were identified via BLASTP analysis (NCBI BLAST+ v2.13.0) against the genomes of *Z. galactanivorans* (NCBI reference sequence: NC_015844.1) and *C. algicola* (NCBI reference sequence: NC_014934.1), respectively. The signal peptide prediction was performed using SignalP 6.0 with default parameters [29]. Wheat arabinoxylan was purchased from Megazyme (Bray, Ireland). The coding sequences of ZgXyn10A and CaXyn10B were codon-optimized for *Escherichia coli* expression and synthesized by GenScript (Nanjing, China).

### 4.2. Cloning and Recombinant Expression

The genes (removing the signal peptide) were cloned into the pET-28a (+) vector (Novagen, Madison, WI, USA) using the NdeI and XhoI restriction sites, generating pET-28a-ZgXyn10A and pET-28a-CaXyn10B. The constructs produced N-terminal His-tagged recombinant protein. The recombinant plasmids were transformed into *E. coli* BL21 (DE3)-competent cells (Thermo Fisher Scientific, Waltham, MA, USA). For the protein expression, cultures were grown in auto-induction medium (1% tryptone,0.5% yeast extract, 25 mM Na_2_HPO_4_, 25 mM KH_2_PO_4_, 50 mM NH_4_Cl, 5 mM Na_2_SO_4_, 0.5% glycerol, 0.05% glucose, 0.2% lactose, 2 mM MgSO_4_) at 37 °C (4 h) followed by 20 °C (20 h) [30]. Cells were harvested by centrifugation (8000× *g*, 10 min, 4 °C) and the pellet was stored at −80 °C.

### 4.3. Protein Purification

The pellets were resuspended in lysis buffer (20 mM Na_2_HPO_4_, 1 M NaCl, pH 7.5). Cell disruption was performed via sonication on ice. The crude lysate was loaded onto an Ni-NTA Sepharose column (Cytiva, Uppsala, Sweden) pre-equilibrated with lysis buffer. After washing with lysis buffer containing 20 mM imidazole, the bound proteins were eluted using a linear imidazole gradient (20–500 mM). Fractions containing the target proteins were pooled and dialyzed against 150 mM NaCl, 20 mM Tris-HCl (pH 7.5) at 4 °C. The protein purity was confirmed by sodium dodecyl sulfate–polyacrylamide gel electrophoresis (12% SDS-PAGE), and the concentrations were quantified using the Bradford assay with bovine serum albumin as the standard [31].

### 4.4. Enzymatic Assays

The xylanase activity was determined using the 3,5-dinitrosalicylic acid (DNS) method [32]. The reactions (200 μL total volume) contained 10 mg/mL wheat arabinoxylan in 100 mM sodium citrate–phosphate buffer (pH 7) and 100 μL appropriately diluted enzyme. After incubation at 40 °C (ZgXyn10A) or 30 °C (CaXyn10B) for 10 min, the reactions were stopped by adding 300 μL DNS reagent and boiling for 5 min. The absorbance at 540 nm was measured. One unit (U) of enzyme activity was defined as the amount releasing 1 μmol of xylose equivalents per minute.

The optimal pH for xylanase activity was determined with 100 mM citrate–phosphate (pH 3–8), glycine–NaOH (pH 9–10), and Na_2_HPO_4_–NaOH (pH 11–12) buffers. The optimal temperature for xylanase activity was determined at temperatures ranging from 10 to 60 °C. The pH or temperature yielding the maximum activity was defined as optimal. The pH stability of the enzyme was incubated in buffers (pH 3–12) at 25 °C for 1 h, followed by pH adjustment to the optimal value prior to activity determination. For the thermostability, the enzyme was evaluated by incubating the enzyme at 20–70 °C for up to 1 h, with the residual activity measured at optimal temperature.

To assess the effect of metal ions, organic solvents and chemical agents, the enzymatic assay was performed with the addition of 5 mM of ZnSO_4_, MgSO_4_, CoCl_2_, NiCl_2_, FeCl_2_, MnCl_2_, CaCl_2_, EDTA, SDS, ethanol, 5% (*v*/*v*) of methanol, acetone, DMSO, isopropanol, 100 mM of urea, ammonium sulfate, 5% (*v*/*v*) of glycerol or 0.5% (*v*/*v*) of β-mercaptoethanol.

The Michaelis–Menten constants (*K*_m_, *V*_max_) were determined using 1–8 mg/mL wheat arabinoxylan. Data were fitted to the Michaelis–Menten curve using GraphPad Prism 9.0. All the enzymatic activity assays were performed in triplicate.

### 4.5. Structural Analysis

Multiple sequence alignments were performed using Clustal Omega (https://www.ebi.ac.uk/jdispatcher/msa/clustalo) (accessed on 24 June 2024). Homology models of ZgXyn10A and CaXyn10B were generated using the SWISS-MODEL server (https://swissmodel.expasy.org/) (accessed on 24 June 2024) [33], with *Cellvibrio mixtus* CmXyn10B (PDB: 1UQY, 49% identity) and *Xanthomonas* sp. XynA (PDB: 4PMU, 64% identity) as templates. The structural models were validated using the ERRAT program in SAVESv6.1 [34]. The crystal structures of the thermostable GH10 xylanases TmxB (PDB: 1VBR) and CoXyn10A (PDB: 5Y3X) were retrieved from the Protein Data Bank. The structural alignments and RMSD calculations were performed using PyMOL (Schrödinger, New York, NY, USA).

### 4.6. Molecular Dynamics (MD) Simulations

The MD simulations were conducted using GROMACS 2022.3 [35] with the CHARMM36m force field [36]. Each enzyme was solvated in a cubic box of SPC216 water, with a minimum 1.0 nm distance from the protein to the box edge. Na⁺/Cl^−^ ions were added to neutralize the system. Energy minimization was performed using the steepest descent algorithm (5000 steps). The systems were equilibrated in NVT (100 ps) and NPT (200 ps) ensembles using the Berendsen thermostat (300 K) and Parrinello–Rahman barostat (1 bar). The production runs (100 ns) used a 2 fs time step, with constraints applied to the bonds via the LINCS algorithm. The simulations for each xylanase molecule were performed at 285 K, 300 K, and 315 K. Specifically, the simulations at 300 K were performed in three replicates. The long-range electrostatics were treated using PME (particle mesh Ewald). The root mean square deviation (RMSD) and root mean square fluctuation (RMSF) were calculated for Cα atoms using GROMACS tools. When calculating the RMSD, the molecular conformation from the tpr file after energy minimization was used as the reference. The flexibility of the loop regions was presented using PyMOL.

## Figures and Tables

**Figure 1 ijms-26-03215-f001:**
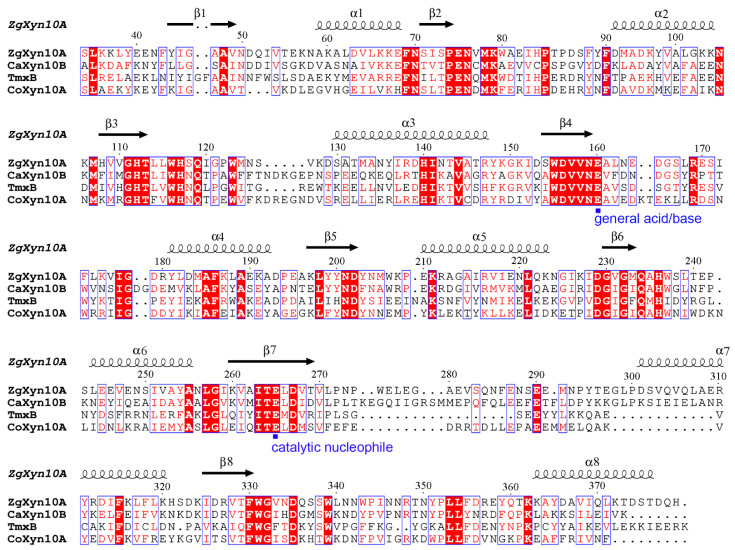
Multiple sequence alignment of the (β/α)_8_ barrel domains from marine and thermophilic xylanases. ZgXyn10A (NCBI accession number: WP_013991550.1) from marine bacterium *Z. galactanivorans*, CaXyn10B (NCBI accession number: WP_013549131.1) from marine bacterium *C. algicola*, TmxB (NCBI accession number: AAD35164.1) from thermophilic *Thermotoga maritima*, and CoXyn10A (NCBI accession number: WP_013411146.1) from thermophilic *Caldicellulosiruptor obsidiansis*. The secondary structure of the ZgXyn10A GH10 domain is indicated above the amino acid sequence (arrows for β-strands and coils for α-helices). The catalytic amino acid residues are marked with blue square symbols.

**Figure 2 ijms-26-03215-f002:**
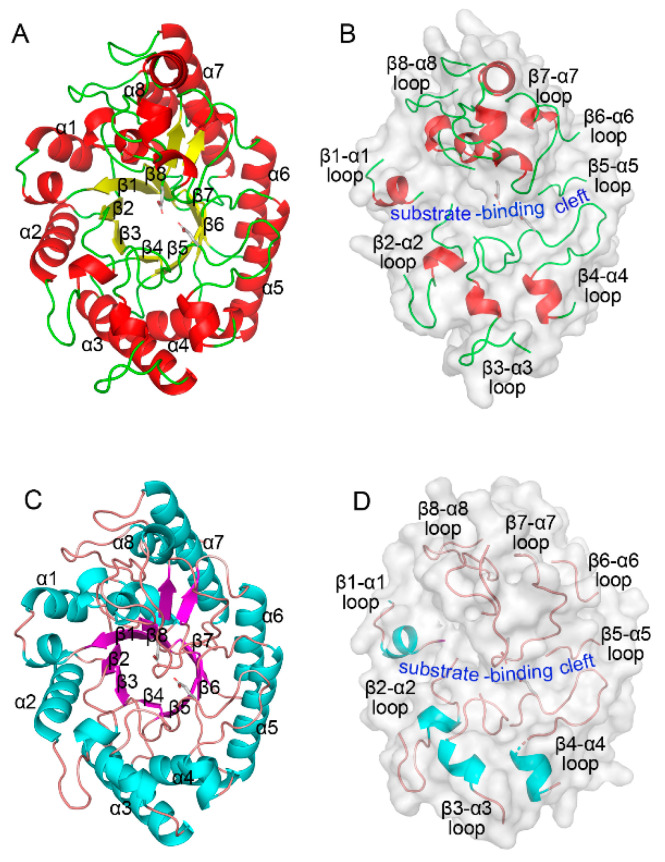
Three-dimensional structural models of the (β/α)_8_ barrel domains in ZgXyn10A and CaXyn10B. (**A**) Cartoon representation of ZgXyn10A, showing the positions and labels of secondary structure elements. The α-helices, β-strands, and loops are colored red, yellow, and green, respectively. (**B**) ZgXyn10A in the same orientation as (**A**), displayed as a surface model with labeled loop regions connecting the α-helices and β-strands, and the location of the substrate-binding cleft. (**C**) Cartoon representation of CaXyn10B, showing the positions and labels of secondary structure elements. The α-helices, β-strands, and loops are colored cyan, purple, and brown, respectively. (**D**) CaXyn10B in the same orientation as (**C**), displayed as a surface model with labeled loop regions and substrate-binding cleft. Catalytic amino acid residues are highlighted as stick models in all the panels.

**Figure 3 ijms-26-03215-f003:**
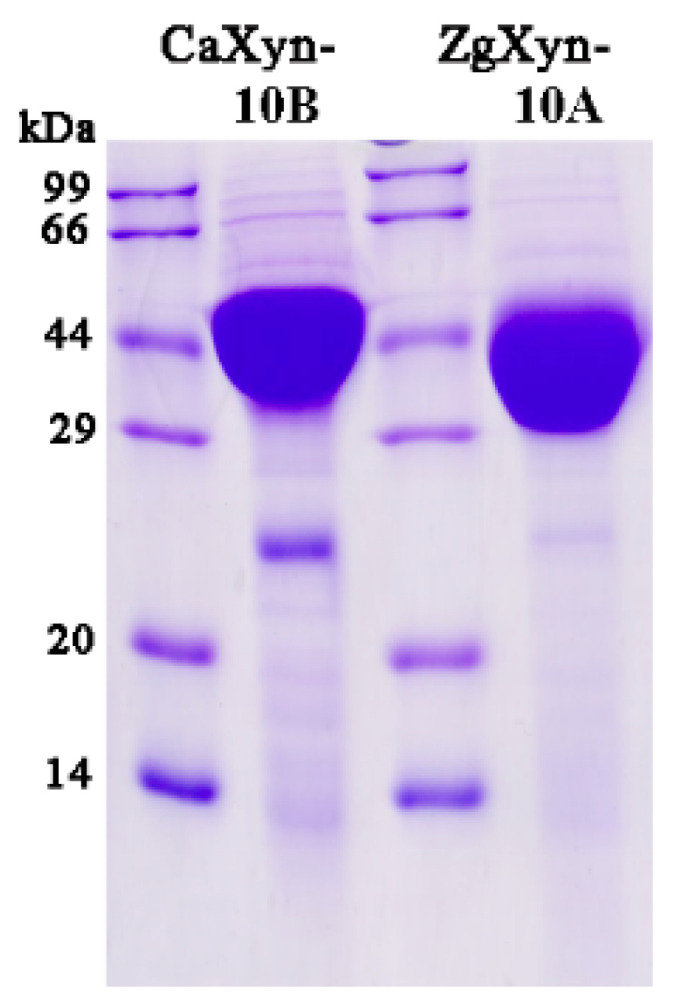
SDS-PAGE (12%) analysis of the purified recombinant xylanases. Soluble fractions of ZgXyn10A (43.7 kDa) and CaXyn10B (44.8 kDa) expressed in *E. coli* BL21 (DE3).

**Figure 4 ijms-26-03215-f004:**
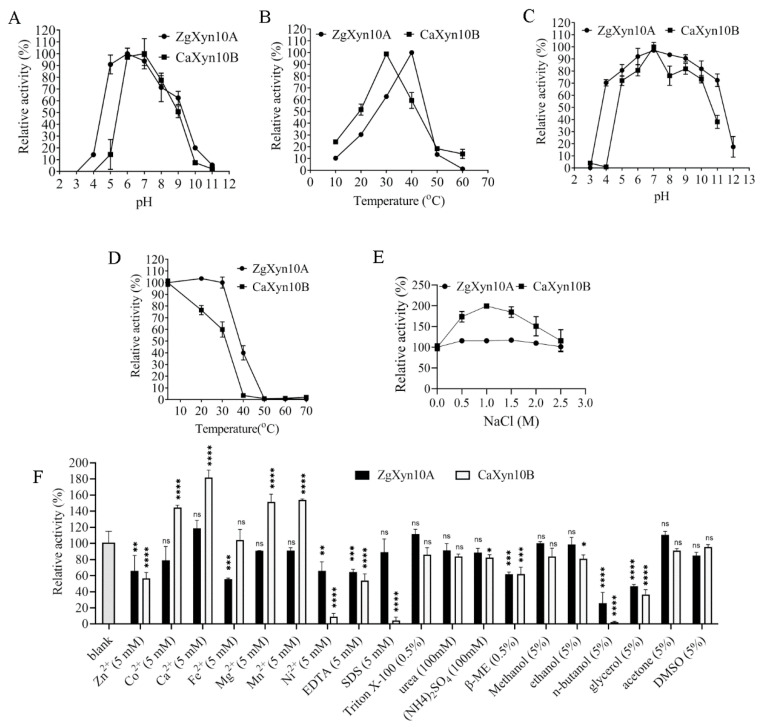
Comparative enzymatic characterization of ZgXyn10A and CaXyn10B. (**A**) pH optima determined at 30 °C using 1% arabinoxylan. (**B**) Temperature optima measured at pH 7.0. (**C**) pH stability assessed after 1 h pre-incubation at 25 °C. (**D**) Thermostability evaluated by residual activity after 1 h incubation at 10–60 °C. (**E**) Halotolerance profile in 0–2.5 M NaCl. (**F**) Effects of metal ions (5 mM) and chemical reagents on relative activity. All the assays were performed in triplicate and the error bars indicate standard deviations. Data represent the mean ± SD. The statistical analysis compared the control group (no additives) with experimental groups containing additives. Significance levels are indicated as follows: ns (not significant), *p* > 0.05; * *p* ≤ 0.05, ** *p* ≤ 0.01, *** *p* ≤ 0.001, **** *p* ≤ 0.0001.

**Figure 5 ijms-26-03215-f005:**
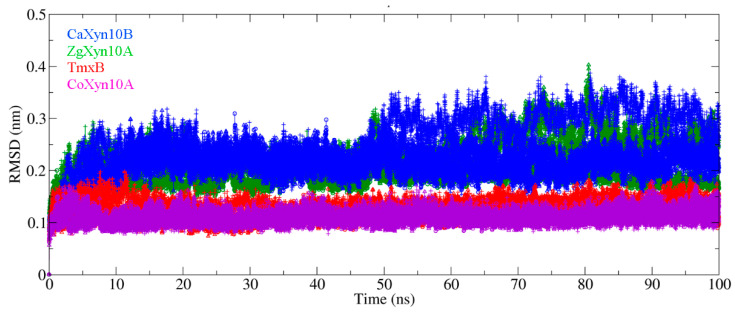
Backbone root mean square deviation (RMSD) trajectories from the 100 ns molecular dynamics simulations at 300 K. The post-minimization structure serves as the RMSD calculation baseline. Each xylanase molecule (ZgXyn10A in green, CaXyn10B in blue, TmxB in red, CoXyn10A in purple) was simulated three times. The replicates for each enzyme are shown as circles, triangles, and plus signs.

**Figure 6 ijms-26-03215-f006:**
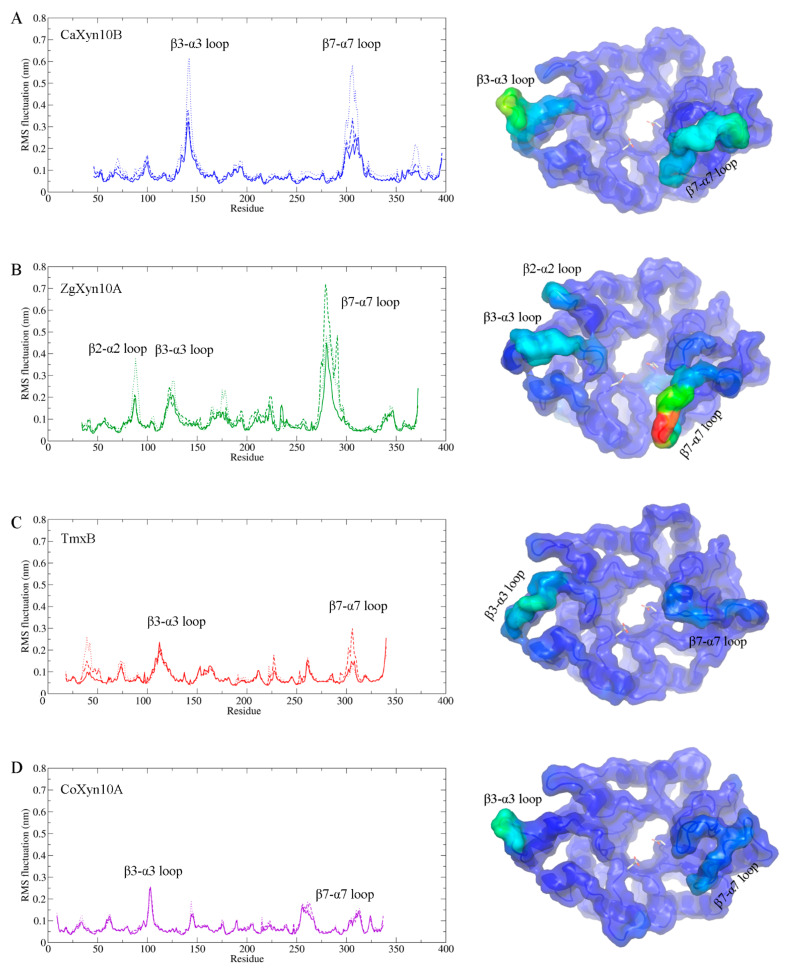
Residue-specific flexibility analysis of marine xylanases. (Left) Root mean square fluctuation (RMSF) profiles of ZgXyn10A (green, (**B**)), CaXyn10B (blue, (**A**)), TmxB (red, (**C**)), and CoXyn10B (purple, (**D**)). The data were derived from simulations at 300 K. Each xylanase molecule was simulated for three replicates, with the results shown as solid lines, long dashed lines, and short dashed lines. (Right) Surface representations colored by RMSF (blue: rigid, red: flexible). Key flexible regions annotated: β3-α3 and β7-α7 loops. Catalytic triads are shown as sticks.

**Table 1 ijms-26-03215-t001:** Amino acid and structural alignments of the GH10 domains of CaXyn10B, ZgXyn10A, TmxB, and CoXyn10A. The values represent the amino acid residue sequence identity and the RMSD values of α-carbon atoms, respectively.

	ZgXyn10A	TmxB	CoXyn10A
CaXyn10B	42%, 1.56 Å	27%, 1.25 Å	35%, 1.67 Å
ZgXyn10A	/	30%, 1.30 Å	33%, 1.78 Å
TmxB	/	/	34%, 1.52 Å

**Table 2 ijms-26-03215-t002:** Kinetic parameters of ZgXyn10A and CaXyn10B.

Enzyme	*V*_max_ (U/mg)	*K*_m_ (mg/mL)	*k*_cat_ (s^−1^)	*k*_cat_/*K*_m_ (mL/s/mg)
CaXyn10B	119.86	16.00	163.69	10.23
ZgXyn10A	1532.11	8.84	1017.65	115.12

## Data Availability

The original contributions presented in this study are included in the article/Appendix A.

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
