# Peer review of "Loop Dynamics Mediate Thermal Adaptation of Two Xylanases from Marine Bacteria"

_ijms, 2025, doi:10.3390/ijms26073215_

Round 1
Reviewer 1 Report
Comments and Suggestions for Authors
Dear Authors
In the manuscript “Loop dynamics mediate thermal adaptation of two xylanases from marine bacteria” the authors suggest that two xylanases from two bacteria Z. galactonovorans and C. algicola have a structural adaptation that allow a thermal behavior. Although the results of this article are relevant, some revision is needed before publication.
Introduction
Change hydrolyzes the b-1,4-xylosidic linkages in xylan, a major component… by hydrolyze 1,4-b-D-xylosidic linkages in xylan, a major structural polysaccharide in plant cells or major component of hemicellulose.
Materials and Methods
Specify how did you removed the signal peptide of the genes?
Section 4.3 Change the correct concentration used in the dialysis buffer. Indicate the percentage of SDS-PAGE used. Include the citation of the Bradford method.
Section 4.4 Based on this information, citrate-phosphate buffer has a maximum pH value of 7 and Tris-HCl it has a pH range of 7-9. https://diverdi.colostate.edu/C431/experiments/determination%20of%20the%20pKa%20of%20a%20weak%20acid/references/meth_in_enzymol_1955_v1_p138.pdf
Specify which buffer is used at pH 12 for pH stability.
Section 4.5 Include validation of homology models, such as Ramachandra or ERRAT.
Section 4.6 Include how did you performed the flexibility analysis.
Include PyMOL v.2.5 (Lilkova et al., 2015, RRID:SCR_000305)
Results
Section 2.1 Include the number of residue of the signal peptide of ZgXyn10A gene.
I suggest that you make the assignment of secondary elements, motifs and domains from one of the PDB sequences used in your study. In this analysis include sequences from non-thermophilic xylanases and determine the difference between them.
Figure 2. Indicate the percentage of SDS-PAGE used.
Figure 3. Remove n=3 biological replicates and replace by All assays were
performed in triplicate and error bars indicate standard deviations.
Table 1. Parenthesis is missing in ml/s/mg.
Add a space between Table 1 and the paragraph that begins “Metal ion effects….”
In the paragraph Metal ion effects should be mentioned as in the Fgure 3.
Revise the phase EDTA inhibition was more pronounced in…
Change the word sensitive to inhibited
Figure 5 I recommend placing the images where the 3D pymol structure with its corresponding cavity (using POCASA software) identifies the main residues. Include a reference where indicated what are the catalytic triad and show the catalytic triad.
In the paragraph that starts with RMSF analysis what is the b2-a2 loop? In the same paragraph, next to the catalytic residues the question is What are these residues? Also place them in the alignment shown.
Discussion
In the four paragraph of this section is poorly worded, was the loop directly correlated with temperature?
The author should discuss more about the loop dynamics as a key evolutionary driver in the thermal adaptation of GH10 xylanases.

In general the manuscript is well written
Author Response
In the manuscript “Loop dynamics mediate thermal adaptation of two xylanases from marine bacteria” the authors suggest that two xylanases from two bacteria Z. galactonovorans and C. algicola have a structural adaptation that allow a thermal behavior. Although the results of this article are relevant, some revision is needed before publication.
Introduction
Comment 1: Change hydrolyzes the b-1,4-xylosidic linkages in xylan, a major component… by hydrolyze 1,4-b-D-xylosidic linkages in xylan, a major structural polysaccharide in plant cells or major component of hemicellulose.
Response 1:We agree that refining the description of the enzymatic action can enhance the clarity and precision of our work. We have revised the relevant section to read: “The enzyme hydrolyzes 1,4-β-D-xylosidic linkages in xylan, a major structural polysaccharide in plant cells or a primary component of hemicellulose.” This change aligns with the biochemical convention for describing glycosidic bonds and better reflects the complexity of the substrate.
Materials and Methods
Comment 2: Specify how did you removed the signal peptide of the genes?
Response 2:The removal of signal peptides from the genes encoding the GH10 xylanases was carried out using standard molecular biology techniques. Specifically, we used bioinformatics tools (SignalP-6.0: https://services.healthtech.dtu.dk/services/SignalP-6.0/) to identify the signal peptide sequences within the amino acid sequences of the enzymes. These tools allowed us to predict the start of the mature protein region. We have addressed the information in Methods.
Comment 3: Section 4.3 Change the correct concentration used in the dialysis buffer. Indicate the percentage of SDS-PAGE used. Include the citation of the Bradford method.
Response 3:We have made the following changes to address your concerns: (1) We have corrected the typographical errors in the composition of the dialysis buffer solution. (2) We have indicated the percentage of SDS-PAGE gels used in our experiments. Specifically, we used 12% SDS-PAGE gels for all protein separation and analysis steps. (3) We have included a citation for Bradford assay in our manuscript (Kruger NJ. The Bradford method for protein quantitation. Methods Mol Biol. 1994;32:9-15).
Comment 4: Section 4.4 Based on this information, citrate-phosphate buffer has a maximum pH value of 7 and Tris-HCl it has a pH range of 7-9. https://diverdi.colostate.edu/C431/experiments/determination%20of%20the%20pKa%20of%20a%20weak%20acid/references/meth_in_enzymol_1955_v1_p138.pdf
Response 4:In Section 4.4, we used 100 mM citrate-phosphate buffer tailored to cover pH 3–8. To ensure buffer capacity at higher pH, we adjusted the molar ratio of disodium phosphate (Na₂HPO₄) to citric acid as described by McIlvaine (MCILVAINE, T.C. A buffer solution for col orimetric comparison. Journal of Biological Chemistry, 1921, 49(1), 183–186, doi:10.1016/ s0021-9258(18)86000-8).
Comment 5: Specify which buffer is used at pH 12 for pH stability.
Response 5:"For pH 12 stability testing, the enzyme was incubated in Na₂HPO₄-NaOH buffer (100 mM, adjusted with 2 M NaOH), followed by activity determination." The revised revision of manuscript resolved the ambiguity.
Comment 6: Section 4.5 Include validation of homology models, such as Ramachandra or ERRAT.
Response 6:Thank you for the reviewer's very professional question. It's true that the three-dimensional structures from homology modeling need to be evaluated for their protein structure quality. We used the ERRAT online program (available at https://saves.mbi.ucla.edu/) to analyze the two structural models discussed in the paper. In the revised manuscript, we have added the corresponding experimental methods, citing the ERRAT reference (Colovos C, Yeates TO. Verification of protein structures: patterns of nonbonded atomic interactions. Protein Sci. 1993 Sep;2(9):1511-9). The results of the ERRAT analysis have been included in the supplementary material.
Comment 7: Section 4.6 Include how did you performed the flexibility analysis.
Response 7:Thank you for pointing out the need for clarification in Section 4.6. We realize there was a mistake in our previous description. We have revised the text to state: "The flexibility of loop regions was presented using PyMOL." Actually, the flexibility analysis of the protein structure was conducted by calculating the Root Mean Square Fluctuation (RMSF) values of each amino acid residue's backbone after molecular dynamics simulations. The RMSF values simulate the temperature factors of the amino acid residues, and in PyMOL, we used warm colors to highlight the structural regions with higher temperature factors.
Comment 8: Include PyMOL v.2.5 (Lilkova et al., 2015, RRID:SCR_000305)
Response 8:We have adopted this suggestion and cited the following reference in the text. Lilkova, E., Petkov, P., Ilieva, N. and Litov, L. (2015) The PyMOL Molecular Graphics System. 10919-25.
Results
Comment 9: Section 2.1 Include the number of residue of the signal peptide of ZgXyn10A gene. I suggest that you make the assignment of secondary elements, motifs and domains from one of the PDB sequences used in your study. In this analysis include sequences from non-thermophilic xylanases and determine the difference between them.
Response 9:In the revised manuscript, the location of the signal peptide for ZgXyn10A has been added to Section 2.1. The original Figure 1 has been remade. The new Figure 1 now includes a secondary structure display. The secondary structure of the ZgXyn10A GH10 domain is indicated above the amino acid sequence (arrows for β-strands and coils for α-helices).
Comment 10: Figure 2. Indicate the percentage of SDS-PAGE used.
Response 10:The percentage of SDS-PAGE used indicated in Figure 2 legend and method section.
Comment 11: Figure 3. Remove n=3 biological replicates and replace by All assays were
performed in triplicate and error bars indicate standard deviations.
Response 11:The modified was made.
Comment 12: Table 1. Parenthesis is missing in ml/s/mg.
Add a space between Table 1 and the paragraph that begins “Metal ion effects….”
Response 12:All the suggestions were performed in the revised manuscript.
Comment 13: In the paragraph Metal ion effects should be mentioned as in the Fgure 3.
Revise the phase EDTA inhibition was more pronounced in…
Change the word sensitive to inhibited
Response 13:All the suggestions were performed in the revised manuscript.
Comment 14: Figure 5 I recommend placing the images where the 3D pymol structure with its corresponding cavity (using POCASA software) identifies the main residues. Include a reference where indicated what are the catalytic triad and show the catalytic triad.
Response 14: The substrate-binding cleft of the GH10 family xylanases is a long groove formed by loops connecting each pair of β-strands and α-helices. This characteristic structure of the family has been shown in many related studies. The reviewer's suggestion is great and helpful for readers who are not familiar with the GH10 family structure. We made following modification in the revised manuscript.
In Figure 1, the positions of the catalytic triads for the four xylanases in our study were highlighted in multiple sequence alignment.
We rewrote the related discussion, focusing on adding information about the GH10 family's substrate-binding cleft, loops, and catalytic triad. A reference GH10 xylanase was included there.
We generated a new figure (Figure 2) presenting the three-dimensional structures of two novel xylanase molecules investigated in this study. The image clearly labels the names of secondary structure elements and indicates the positions of active site clefts on the enzyme molecules.
Comment 15: In the paragraph that starts with RMSF analysis what is the b2-a2 loop? In the same paragraph, next to the catalytic residues the question is What are these residues? Also place them in the alignment shown.
Response 15:we previously did not explicitly label structural components of GH10 family xylanases. We have now generated Figure 2, which clearly illustrates the positions of each structural loop and catalytic amino acid residues. The specific names of these catalytic residues are provided in this paragraph.
Discussion
Comment 16: In the four paragraph of this section is poorly worded, was the loop directly correlated with temperature?
Response 16:We appreciate the reviewer's astute observation. We fully concur with this critical point. The entire Discussion section has been revised, with particular emphasis on avoiding definitive claims regarding the relationship between loop regions and thermostability characteristics in GH10 family xylanases. Instead, we now emphasize that while we have observed this correlation, it serves as an empirical basis for preliminary predictions of enzymatic temperature preferences. Future analyses will require examining additional characterized xylanases to validate this relationship.
Comment 17: The author should discuss more about the loop dynamics as a key evolutionary driver in the thermal adaptation of GH10 xylanases.
Response 17:We revised the Discussion section to provide a clearer analysis of the correlation between loop regions and temperature-dependent characteristics.
Reviewer 2 Report
Comments and Suggestions for Authors
The authors present two newly identified xylanases, ZgXyn10A and CaXyn10B, with experimental evidences on their activity and in silico analyses comparing their dynamics to two thermostable xylanases. It is very interesting that their optimal temperatures are lower than of the thermostable xylanases and that their sequence similarity are also low. But the in silico analyses do not go beyond the usual minimum, even lack replicates for consistency check. And the conclusion about the identification of the loops does not bring novelty, since the loops of these proteins are already known to be the sources of flexibility and key to their function. It would be more interesting if the authors would explore the sources for their structural conservation, despite the low sequence identity, and search for structural and dynamic reasons for the differences in optimal temperature - which residues are responsible for this loop being more or less flexible?
As it is, this is a simple study with important findings - the description of these two newly found xylanases. The conclusions could better highlight these novelties instead of superficially discussing the dynamics of the loops. I recommend the following revisions.
Major revisions:
- Please perform two more replicates of the molecular dynamics.
- In the discussion, it is not clear how the results validate homology modelling
- The predictive framework proposed in the discussion is not obvious. Please explain it in more details.
- It is also not clear why only 10 ns should be enough for this predictive framework, specially without replicates in this study.
- Please add to the discussion that 0.15 and 0.25 Å are usually considered very low. And that this result is consistent with the core of these proteins being very rigid, and the differences in the RMSD evince the dynamics of the loops. It is interesting to me that the closer to the optimal temperature, the higher the RMSD values obtained in the MDs. Was this expected? Can you comment on that?
- Was the conclusion that “halotolerance evolves independently from thermal resilience” drawn from the presented results? If so, this conclusion is not obvious. Please explain more. If not, please give a reference.
Minor revisions:
- Identities and RMSDs in results 2.1 would be better in two small tables or colored matrices.
- In figure 1, please indicate the barrel domain and the catalytical domain as two regions in the MSA.
- In figure 1, what does structural superimposition mean? Were the catalytic domains aligned in a different way other than with Clustal?
- Please avoid adjectives that decrease the precision of a result, for example the first sentence of results 2.2 would be better like “Both enzymes were expressed in coli and purified to high concentrations.”
- Can you mention and give a reason to the increase in activity of CaXyn10B with higher salt concentration showed in figure 3e?
- Can you briefly introduce the need to test the activity in so many conditions, including the presence of many different ions and small molecules?
- Please clarify if the RMSD was calculated comparing each frame with the first, or each frame with its previous, or each frame with a mean conformation. Preferably, it should be compared with a mean conformation. If it was compared with the first conformation, it might be that the higher values for ZgXyn10A and CaXyn10B indicate only that the starting conformation was not well minimized or equilibrated, or even that the models were farther from a native conformation.
- In the third paragraph of the discussion, correct “Compare to ZgXyn10A...” to “Compared with ZgXyn10A”.
- The last sentence of third paragraph of the discussion is not clear. Please rephrase it.
- Can you give a reference to the last sentence of the fourth paragraph of the discussion?
- Refrain from discussing results not shown as in the first sentence of the fifth and sixth paragraphs.
- Please correct the typo in “This observation provided a computational approach ...” to “This observation provides a computational approach ...”
- The supplementary material is not mentioned in the text.
Author Response
The authors present two newly identified xylanases, ZgXyn10A and CaXyn10B, with experimental evidences on their activity and in silico analyses comparing their dynamics to two thermostable xylanases. It is very interesting that their optimal temperatures are lower than of the thermostable xylanases and that their sequence similarity are also low. But the in silico analyses do not go beyond the usual minimum, even lack replicates for consistency check. And the conclusion about the identification of the loops does not bring novelty, since the loops of these proteins are already known to be the sources of flexibility and key to their function. It would be more interesting if the authors would explore the sources for their structural conservation, despite the low sequence identity, and search for structural and dynamic reasons for the differences in optimal temperature - which residues are responsible for this loop being more or less flexible?
As it is, this is a simple study with important findings - the description of these two newly found xylanases. The conclusions could better highlight these novelties instead of superficially discussing the dynamics of the loops. I recommend the following revisions.
Major revisions:
Comment 1: Please perform two more replicates of the molecular dynamics.
Response 1: We conducted three replicate molecular dynamics (MD) simulations at 300K (room temperature) as suggested by the reviewer. The corresponding results have been compiled and presented in the Methods and Results sections.
Comment 2: In the discussion, it is not clear how the results validate homology modelling
Response 2: The structural models were validated using ERRAT program in SAVESv6.1 (Supplementary Figure S1). Following reference was added in the revised manuscript.
Comment 3: The predictive framework proposed in the discussion is not obvious. Please explain it in more details.
Response 3: Due to concerns raised by another reviewer regarding the adequacy of our proposed framework for predicting thermostability characteristics of GH10 family xylanases via molecular dynamics simulations, we have revised the manuscript. Specifically, we removed the detailed simulation parameters presented in the initial draft. Instead, we now emphasize the observed correlation between loop flexibility and temperature-dependent properties, and propose that this relationship offers an empirical method for predicting thermostability features within GH10 family xylanases.
Comment 4: It is also not clear why only 10 ns should be enough for this predictive framework, specially without replicates in this study.
Response 4: We fully concur with the reviewer's critique. During revisions, we conducted three 100-ns molecular dynamics simulations for each xylanase molecule. The results demonstrate that the previous 10-ns simulation duration was insufficient, and performing three replicates was necessary.
Comment 5: Please add to the discussion that 0.15 and 0.25 Å are usually considered very low. And that this result is consistent with the core of these proteins being very rigid, and the differences in the RMSD evince the dynamics of the loops. It is interesting to me that the closer to the optimal temperature, the higher the RMSD values obtained in the MDs. Was this expected? Can you comment on that?
Response 5: We added relevant discussions in the Discussion section, for example: "Although a difference of 0.1-0.2 nm in length is not considered significant in protein three-dimensional structural comparisons, all four xylanase molecules consistently maintained this disparity throughout nearly the entire 100 ns simulation period."
The molecular dynamics simulations in this study were performed at three temperatures: 285K, 300K, and 315K (approximately 10°C, 25°C, and 40°C, respectively). Figure 5 presents the simulation results at 25°C, which is closest to the optimal temperature of CaXyn10B (30°C). RMSD analysis showed that CaXyn10B exhibited the highest RMSD values. We attribute this to its lower optimal temperature, as our findings indicate a trend where lower temperatures correlate with greater enzymatic flexibility.
Comment 6: Was the conclusion that “halotolerance evolves independently from thermal resilience” drawn from the presented results? If so, this conclusion is not obvious. Please explain more. If not, please give a reference.
Response 6: We removed the sentence discussing the relationship between salt tolerance and structure in GH10 xylanases, as it was contextually irrelevant within the section focused on temperature characteristics and structural analysis. The article does not delve into salt tolerance, and our research findings show minimal correlation with this property.
Minor revisions:
Comment 7: Identities and RMSDs in results 2.1 would be better in two small tables or colored matrices.
Response 7: In the revised manuscript, the results in Section 2.1 have been compiled into Table 1. The textual description of the results has also been simplified.
Comment 8: In figure 1, please indicate the barrel domain and the catalytical domain as two regions in the MSA.
Response 8: In the initial manuscript, the domain used for sequence and structural alignments was not clearly defined. The revised manuscript clarifies that the (β/α)8 barrel domain is the catalytic domain. To enhance clarity, the revision provides a more detailed description of this domain's secondary structural elements and the positions of catalytic amino acids, and includes Figure 2 to visually present the structure of the (β/α)8 barrel domain.
Comment 9: In figure 1, what does structural superimposition mean? Were the catalytic domains aligned in a different way other than with Clustal?
Response 9: Structural superimposition was performed in PyMOL using its align function to calculate the average deviation of α-carbon atoms after maximizing structural overlap, not in Clustal Omega.
Comment 10: Please avoid adjectives that decrease the precision of a result, for example the first sentence of results 2.2 would be better like “Both enzymes were expressed in coli and purified to high concentrations.”
Response 10: The suggestion was applied in the revised manuscript.
Comment 11: Can you mention and give a reason to the increase in activity of CaXyn10B with higher salt concentration showed in figure 3e?
Response 11: Marine-derived microbial enzymes often exhibit strong salt tolerance. For example, several GH10 family xylanases demonstrate this trait. Typically, these enzymes achieve salt resistance through their surface-rich acidic amino acids (like aspartic acid and glutamic acid). The negatively charged residues form dynamic salt bridges with sodium (Na⁺) or chloride (Cl⁻) ions in high-salt conditions, protecting the enzyme structure from damage.
In the case of CaXyn10B, salt ions (such as Na⁺ or K⁺) might act as allosteric effectors. By binding to non-catalytic sites, they induce conformational changes that enhance enzymatic activity. This interaction explains the observed salt-activated phenomenon.
Comment 12: Can you briefly introduce the need to test the activity in so many conditions, including the presence of many different ions and small molecules?
Response 12: The effects of metal ions, organic solvents, and detergents on enzyme activity are often studied to provide guidance for their practical use in industrial catalysis applications.
Comment 13: Please clarify if the RMSD was calculated comparing each frame with the first, or each frame with its previous, or each frame with a mean conformation. Preferably, it should be compared with a mean conformation. If it was compared with the first conformation, it might be that the higher values for ZgXyn10A and CaXyn10B indicate only that the starting conformation was not well minimized or equilibrated, or even that the models were farther from a native conformation.
Response 13: When calculating RMSD, we use the molecular conformation from the tpr file after energy minimization as the reference. This represents the enzyme's starting conformation, and it is valid to use the post-minimization structure as the RMSD calculation baseline.
Comment 14: In the third paragraph of the discussion, correct “Compare to ZgXyn10A...” to “Compared with ZgXyn10A”.
Response 14: The modification has been made.
Comment 15: The last sentence of third paragraph of the discussion is not clear. Please rephrase it.
Response 15: We have rephrased the sentence as follows: However, unlike typical cold-active enzymes that maintain a significantly higher percentage of their optimal activity between 0-10°C, both enzymes studied here showed reduced performance. They retained only 10-50% residual activity even at the relatively higher temperature range of 10-20°C (Figure 4)
Comment 16: Can you give a reference to the last sentence of the fourth paragraph of the discussion?
Response 16: We included the following reference: Chong W, Zhang Z, Li Z, Meng S, Nian B, Hu Y. Hook loop dynamics engineering transcended the barrier of activity-stability trade-off and boosted the thermostability of enzymes. Int J Biol Macromol. 2024, 278(Pt 4):134953.
Comment 17: Refrain from discussing results not shown as in the first sentence of the fifth and sixth paragraphs.
Response 17: In the revised manuscript, descriptions of “results not shown” have been removed. The mentioned results are now displayed as supplementary figures in the supplementary materials.
Comment 18: Please correct the typo in “This observation provided a computational approach ...” to “This observation provides a computational approach ...”
Response 18: The correction has been made.
Comment 19: The supplementary material is not mentioned in the text.
Response 19: We have included the supplementary material in the revised manuscript.
Reviewer 3 Report
Comments and Suggestions for Authors
In this manuscript, the authors investigate GH10 xylanases extracted from Zobellia Galactanovorans and Cellulophaga algicola, namely ZgXyn10A, CaXyn10B, which show reduced activity at temperatures below 30 C. They also consider two thermostable (thermophilic) marine GH10 xylanases, namely TmxB, CoXyn10A. Then, they show that albeit the four xylaneses are structurally very similar because they have similar aminoacid sequences, there are differences in their thermal stability as indicated by loop flexibility and substrate-binding cleft dynamics. To corroborate their claim, they show that the RMSD and beta7-alpha7 loop root mean square peak fluctuations, as measured in 100 ns long molecular dynamics (MD) simulations, exhibit lower values for thermostable xylanases. They also claim that thermal stability seems loosely affected by hydrogen bonding, but do not report any data in the manuscript. I believe there are insufficient data shown to support the claim that this approach provides a general framework to predict thermal behavior of xylanases. I would suggest that the authors perform additional simulations at different temperatures, which seems relevant since they would like to unveil the reason why some families of xylanases are thermostable, and some others are not. What is the temperature chosen in their simulations?
Comments on the Quality of English LanguageI would suggest that English is revised throughout the manuscript because some sentences are very cryptic and not really supported enough by the data reported in the manuscript.
Author Response
In this manuscript, the authors investigate GH10 xylanases extracted from Zobellia Galactanovorans and Cellulophaga algicola, namely ZgXyn10A, CaXyn10B, which show reduced activity at temperatures below 30 C. They also consider two thermostable (thermophilic) marine GH10 xylanases, namely TmxB, CoXyn10A. Then, they show that albeit the four xylaneses are structurally very similar because they have similar aminoacid sequences, there are differences in their thermal stability as indicated by loop flexibility and substrate-binding cleft dynamics. To corroborate their claim, they show that the RMSD and beta7-alpha7 loop root mean square peak fluctuations, as measured in 100 ns long molecular dynamics (MD) simulations, exhibit lower values for thermostable xylanases. They also claim that thermal stability seems loosely affected by hydrogen bonding, but do not report any data in the manuscript.
Comment: I believe there are insufficient data shown to support the claim that this approach provides a general framework to predict thermal behavior of xylanases. I would suggest that the authors perform additional simulations at different temperatures, which seems relevant since they would like to unveil the reason why some families of xylanases are thermostable, and some others are not. What is the temperature chosen in their simulations?
Response: We thank the reviewer for raising this critical point. All initial MD simulations were conducted at 300 K (room temperature) because this aligns with the experimentally determined optimal activity range (Figure 4A) of ZgXyn10A and CaXyn10B. This design ensures biological relevance while exploring structural dynamics.
To confirm the reliability of our findings, additional simulations at 325K and 285K were performed for all four xylanases during manuscript revision (Supplementary Materials). The results showed temperature differences (15°C intervals) significantly affected the cold-adapted enzymes ZgXyn10A and CaXyn10B. Specifically, ZgXyn10A exhibited increasing RMSD values with rising temperatures, while CaXyn10B showed no consistent trends. In contrast, the thermophilic enzymes TmxB and CoXyn10A remained unaffected across the 285K-315K temperature range.
Reviewer 4 Report
Comments and Suggestions for Authors
The authors present the biochemical characterization of two family GH10 xylanases from marine bacteria. These were obtained by searching for xylanases in the genomes of Zobellia galactanivorans and Cellulophaga algicola. The genes were optimized for expression in E. coli, purified, and their activities assayed at different temperatures, pH, NaCl concentration, metals, and chemical reagents to determine their suitability for biotechnological applications. In addition, 100-ns MD simulations were carried out for these two xylanases and two thermostable xylanases, looking for molecular characteristics associated to the activity profiles at different optimal temperatures. This is traced to the dynamics of two loops in the characteristic beta-barrel structure.
My suggestions to the authors are listed below:
- In Figure 1, include a matrix with the % identities among the four proteins; that is easier to follow than the listing in the text. Given that this matrix is symmetrical, the top part can be dedicated to % identity and the lower part to Calpha RMSD.
- Regarding Calpha RMSD, please state whether these values are calculated for the whole proteins or just for the regions that can be superimposed.
- Regarding the sequence alignment, please decorate it with the secondary structure assignment (so the beta-alpha repeats and the location of the important loops are indicated) and with the active site residues. Also indicate whether these sequences contain the signal peptides or correspond to the mature proteins. The NCBI accession numbers for the thermophilic sequences should also be stated.
- The gel in Figure 2 has a prominent band underneath the main band for CaXyn10B. That looks like a contaminating band. There is an equivalent, less intense, band for ZgXyn10A. Therefore, the claim for no significant contaminating bands is suspect, at least for CaXyn10B.
- Regarding Table 1, I strongly suggest the authors to fit their kinetic data directly to the hyperbolic Michaelis-Menten equation, rather than the linear Lineweaver-Burk equation. It is amply documented in basic Biochemistry textbooks (such as Nelson and Cox Lehninger’s Principles of Biochemistry) that the linear version mistreats the original data. GraphPad Prism 9.0 can do the fit to the original equation.
- Regarding the MD simulations, it is standard practice to perform at least three replicas of the simulations (same as for any experiment). The conclusions derived from just one run for each system are not robust.
- In Figure 5, I suggest including in the molecular structures the active site residues, so the relevance of the loops with increased mobility stands out. Relating this to the sequence alignment can be useful, as a recurrent strategy for thermal stability is to shrink long loops. If these two loops (beta3-alpha3 and beta7-alpha7) are shorter in the thermophilic enzymes compared to the two novel enzymes described in this work, that makes sense.
- The fact that the total number of H-bonds does not change significantly does not mean that they are not important. A regional analysis could yield important insights into areas that are needed for global stability and others that are selectively stabilized for the thermophilic enzymes. The nice thing about MD simulations is that these regional analysis are easy to carry out.
- The discussion mentions the conservation of critical residues for function; these should be highlighted in Figure 1.
- Please proofread the discussion section, as there are a few sentences with grammar errors and a mention of 10-ns MD simulations (instead of 100-ns).
- In section 4.2, please spell out ZY and NPS. Also, indicate the location of the His-tag (N-ter or C-ter) in the recombinant proteins.
- In section 4.4, indicate how many replicas of the experiments were carried out, and perform a statistical analysis of all conditions where differences between the two enzymes are claimed.
- In section 4.5, please correct the PDB-ID for TmxB: it is 1VBR, not 1UBR (it took me a while to find it). Regarding the homology models for ZgXyn10A and CaXyn10B, indicate the quality of the models with the parameters given by SWISS-Model.
- In section 4.6, indicate the NaCl concentration used in the simulations; one can have a charge-neutral system with added salt or without added salt.
- I insist that at least three MD replicas for each system should be carried out to yield robust results.
Please proofread the manuscript. In general it is OK, but the discussion section has both typos and grammar mistakes.
Author Response
The authors present the biochemical characterization of two family GH10 xylanases from marine bacteria. These were obtained by searching for xylanases in the genomes of Zobellia galactanivorans and Cellulophaga algicola. The genes were optimized for expression in E. coli, purified, and their activities assayed at different temperatures, pH, NaCl concentration, metals, and chemical reagents to determine their suitability for biotechnological applications. In addition, 100-ns MD simulations were carried out for these two xylanases and two thermostable xylanases, looking for molecular characteristics associated to the activity profiles at different optimal temperatures. This is traced to the dynamics of two loops in the characteristic beta-barrel structure.
My suggestions to the authors are listed below:
Comment 1:In Figure 1, include a matrix with the % identities among the four proteins; that is easier to follow than the listing in the text. Given that this matrix is symmetrical, the top part can be dedicated to % identity and the lower part to Calpha RMSD.
Response 1:Thank you for the reviewers' suggestions. The results in this section have been organized into Table 1, and the related descriptions in the text have been simplified.
Comment 2:Regarding Calpha RMSD, please state whether these values are calculated for the whole proteins or just for the regions that can be superimposed.
Response 2:Cα RMSD calculations only involved the (β/α)8 barrel domain. This was also the region used for multiple sequence alignment of the amino acid sequences.
Comment 3:Regarding the sequence alignment, please decorate it with the secondary structure assignment (so the beta-alpha repeats and the location of the important loops are indicated) and with the active site residues. Also indicate whether these sequences contain the signal peptides or correspond to the mature proteins. The NCBI accession numbers for the thermophilic sequences should also be stated.
Response 3:In the revised manuscript, Figure 1 has been redesigned to include secondary structure information and highlight catalytic amino acid residue positions. Since the multiple sequence alignment shows residue numbering excluding the signal peptide sequence, this region is not displayed in the figure. Additionally, NCBI accession numbers for each sequence have been added to the figure legend for clarity.
Comment 4:The gel in Figure 2 has a prominent band underneath the main band for CaXyn10B. That looks like a contaminating band. There is an equivalent, less intense, band for ZgXyn10A. Therefore, the claim for no significant contaminating bands is suspect, at least for CaXyn10B.
Response 4:We acknowledge the reviewers' comments and have removed the following statement from the revised manuscript: "No significant contaminating bands were detected, confirming the high purity of the preparations used for subsequent enzymatic characterization."
Comment 5:Regarding Table 1, I strongly suggest the authors to fit their kinetic data directly to the hyperbolic Michaelis-Menten equation, rather than the linear Lineweaver-Burk equation. It is amply documented in basic Biochemistry textbooks (such as Nelson and Cox Lehninger’s Principles of Biochemistry) that the linear version mistreats the original data. GraphPad Prism 9.0 can do the fit to the original equation.
Response 5:In the revised manuscript, we analyzed the enzymatic kinetic data using the Michaelis-Menten equation. The new experimental methods and parameters have been included in the text, with the simulated curves based on the Michaelis-Menten equation provided as Supplementary Materials.
Comment 6:Regarding the MD simulations, it is standard practice to perform at least three replicas of the simulations (same as for any experiment). The conclusions derived from just one run for each system are not robust.
Response 6:In the revised manuscript, we have used the results from three molecular dynamics simulations.
Comment 7:In Figure 5, I suggest including in the molecular structures the active site residues, so the relevance of the loops with increased mobility stands out. Relating this to the sequence alignment can be useful, as a recurrent strategy for thermal stability is to shrink long loops. If these two loops (beta3-alpha3 and beta7-alpha7) are shorter in the thermophilic enzymes compared to the two novel enzymes described in this work, that makes sense.
Response 7:Thank you for your constructive feedback. In the revised manuscript, we have linked the multiple sequence alignment results to Figure 5 for discussion. The original Figure 5 is now renumbered as Figure 6. Both Figure 6 and the multiple sequence alignment (Figure 1) now include additional information about catalytic amino acid residues.
Comment 8:The fact that the total number of H-bonds does not change significantly does not mean that they are not important. A regional analysis could yield important insights into areas that are needed for global stability and others that are selectively stabilized for the thermophilic enzymes. The nice thing about MD simulations is that these regional analysis are easy to carry out.
Response 8:We have included the hydrogen bond analysis results in the Supplementary Materials. Examining local hydrogen bonds, such as monitoring hydrogen bond counts in loop regions during simulations, would provide valuable insights. However, our current technical limitations prevent us from addressing this specific aspect.
Comment 9:The discussion mentions the conservation of critical residues for function; these should be highlighted in Figure 1.
Response 9:The catalytic amino acid residues of GH10 xylanase are now highlighted in the revised Figure 1.
Comment 10:Please proofread the discussion section, as there are a few sentences with grammar errors and a mention of 10-ns MD simulations (instead of 100-ns).
Response 10:We have completely rewritten the entire discussion section, correcting both visible errors and those identified by reviewers.
Comment 11:In section 4.2, please spell out ZY and NPS. Also, indicate the location of the His-tag (N-ter or C-ter) in the recombinant proteins.
Response 11:In the revised manuscript, the Experimental Methods section now details the specific components of the culture media. Additionally, we have included comprehensive construction details of the recombinant plasmids to clarify the positioning of the His-tag on the recombinant protein.
Comment 12:In section 4.4, indicate how many replicas of the experiments were carried out, and perform a statistical analysis of all conditions where differences between the two enzymes are claimed.
Response 12:The replicas of enzymatic assays were indicated in the methods. Statistical analysis has been added to Figure 4.
Comment 13:In section 4.5, please correct the PDB-ID for TmxB: it is 1VBR, not 1UBR (it took me a while to find it). Regarding the homology models for ZgXyn10A and CaXyn10B, indicate the quality of the models with the parameters given by SWISS-Model.
Response 13:The PDB entry has been corrected. The quality of two homolog models were indicated by Global Model Quality Estimate value provided by SWISS-MODEL. The models also were validated by ERRAT server. The information was presented in the supplementary material.
Comment 14:In section 4.6, indicate the NaCl concentration used in the simulations; one can have a charge-neutral system with added salt or without added salt.
Response 14:In this study, sodium and chloride ions were added to the system only to establish neutral system, with no additional ions being introduced.
Comment 15:I insist that at least three MD replicas for each system should be carried out to yield robust results.
Response 15:During the revision process, the 300 K simulations were conducted in triplicate, and the corresponding results have been added to the manuscript.
Round 2
Reviewer 1 Report
Comments and Suggestions for Authors
Dear authors
Thank you for attending all my suggestions.
Author Response
We thank the reviewer for all their suggestions, which were also very helpful to us.
Reviewer 2 Report
Comments and Suggestions for Authors
I appreciate the authors considering all my suggestions. The manuscript has improved substantially, still I suggest changing few minor points before being ready to submission:
- In section 4.5 Structural Analysis, please mention that the reference conformation for calculating the RMSD was the minimized starting conformation. Ideally mention this in the legend of figure 5 as well.
- The addition of more simulations at different temperatures do improve the manuscript, but the following two statements, at sections 2.3 and 3, respectively, would need to be confirmed with replicates.
“At three simulated temperatures (285K, 300K, and 315K), the backbone RMSD of ZgXyn10A exhibited a clear temperature-dependent increase, while CaXyn10B showed inconsistent RMSD patterns across temperature conditions (Figure S3)”. This statement implies there is a correlation between temperature and the increase in the RMSD, but the increase in RMSD should be confirmed with replicates and the correlation is not clear since the RMSD at 300K is lower than at 275K. Maybe change this sentence to: “During simulations at different temperatures, the backbone RMSD of ZgXyn10A stabilized at different values (being the highest RMSD at 315K), while that of CaXyn10B fluctuated around the same value for all the three temperatures”.
“As temperature increases, the flexibility of these loops in ZgXyn10A and CaXyn10B exhibited a slight upward trend (Figure S4)”. Though this is true for the analysed simulations, the temperature increase does not seem to be larger than the error exhibited in the comparison of the three replicates at 300K. Still, a real trend, even small, could be confirmed with replicates.
- This sentence in the legend of Figure 6 seems to be missing a word: “Each xylanase molecule was simulated three replicates”
Author Response
I appreciate the authors considering all my suggestions. The manuscript has improved substantially, still I suggest changing few minor points before being ready to submission:
Comment 1:In section 4.5 Structural Analysis, please mention that the reference conformation for calculating the RMSD was the minimized starting conformation. Ideally mention this in the legend of figure 5 as well.
Response 1:Thank you for the reviewer's suggestion. This important method was previously only explained in the response to reviewers, not included in the article. We have now added this content during revisions.
Comment 2:The addition of more simulations at different temperatures do improve the manuscript, but the following two statements, at sections 2.3 and 3, respectively, would need to be confirmed with replicates.
“At three simulated temperatures (285K, 300K, and 315K), the backbone RMSD of ZgXyn10A exhibited a clear temperature-dependent increase, while CaXyn10B showed inconsistent RMSD patterns across temperature conditions (Figure S3)”. This statement implies there is a correlation between temperature and the increase in the RMSD, but the increase in RMSD should be confirmed with replicates and the correlation is not clear since the RMSD at 300K is lower than at 275K. Maybe change this sentence to: “During simulations at different temperatures, the backbone RMSD of ZgXyn10A stabilized at different values (being the highest RMSD at 315K), while that of CaXyn10B fluctuated around the same value for all the three temperatures”.
“As temperature increases, the flexibility of these loops in ZgXyn10A and CaXyn10B exhibited a slight upward trend (Figure S4)”. Though this is true for the analysed simulations, the temperature increase does not seem to be larger than the error exhibited in the comparison of the three replicates at 300K. Still, a real trend, even small, could be confirmed with replicates.
Response 2: It should be added to Section 4.6 and the legend of Figure 5. We agree with the reviewer's suggestion and made corrections in both places.
Comment 3:This sentence in the legend of Figure 6 seems to be missing a word: “Each xylanase molecule was simulated three replicates”
Response 3:We changed the sentence to "Each xylanase molecule was simulated for three replicates."
Reviewer 3 Report
Comments and Suggestions for Authors
I believe that the manuscript could be published now that the additional data are reported in the supplementary file.
Author Response

(The authors gave the same response as above.)
